# Opening the digital doorway to sexual healthcare: Recommendations from a behaviour change wheel analysis of barriers and facilitators to seeking online sexual health information and support among underserved populations

**Julie McLeod** [1]*, **Claudia S. Estcourt**[1], **Jennifer MacDonald**[1], **Jo Gibbs**[2], **Melvina Woode Owusu**[2], **Fiona Mapp**[2], **Nuria Gallego Marquez** [2], **Amelia McInnes-Dean**[2], **John M. Saunders**[2,3], **Ann Blandford** [4], **Paul Flowers**[5]

1 School of Health and Life Sciences, Glasgow Caledonian University, Glasgow, Scotland, United Kingdom, 2 Institute for Global Health, University College London, London, England, United Kingdom, 3 UK Health Security Agency (UKHSA), London, England, United Kingdom, 4 UCL Interaction Centre (UCLIC), University College London, London, England, United Kingdom, 5 Psychological Science and Health, University of Strathclyde, Glasgow, Scotland, United Kingdom

* julie.mcleod@gcu.ac.uk

## Abstract

### Background

The ability to access and navigate online sexual health information and support is increasingly needed in order to engage with wider sexual healthcare. However, people from underserved populations may struggle to pass though this "digital doorway". Therefore, using a behavioural science approach, we first aimed to identify barriers and facilitators to i) seeking online sexual health information and ii) seeking online sexual health support. Subsequently, we aimed to generate theory-informed recommendations to improve these access points.

### Methods

The PROGRESSPlus framework guided purposive recruitment (15.10.21–18.03.22) of 35 UK participants from diverse backgrounds, including 51% from the most deprived areas and 26% from minoritised ethnic groups. Using semi-structured interviews and thematic analysis, we identified barriers and facilitators to seeking online sexual health information and support. A Behaviour Change Wheel (BCW) analysis then identified recommendations to better meet the needs of underserved populations.

### Results

We found diverse barriers and facilitators. Barriers included low awareness of and familiarity with online information and support; perceptions that online information and support were unlikely to meet the needs of underserved populations; overwhelming volume of information sources; lack of personal relevancy; chatbots/automated responses; and response wait

**Data Availability Statement:** Due to the sensitive nature of the questions asked in this study, participants were assured that transcripts would remain private and confidential and would not be shared beyond the use of anonymised illustrative quotes in publications about the research. In line with our ethical approvals form Glasgow Caledonian University (GCU) Nursing and Community Health Research Ethics Committee (HLS/NCH/20/045) and the East of England - Cambridge South Research Ethics Committee (21/EE/0148), participants were not asked and did not consent to sharing their full transcripts. The transcripts will be stored long-term, for a minimum of 10 years (from 2021) on a secure GCU network drive. Data access requests may be made to the corresponding author, Julie McLeod (julie.mcleod@gcu.ac.uk), the SEQUENCE digital team (cnwl.sequencedigital@nhs.net) or the Glasgow Caledonian University Research Ethics Committee (hlsethics@gcu.ac.uk) and the East of England - Cambridge South Research Ethics Committee (cambridgesouth.rec@hra.nhs.uk). All Behaviour Change Wheel analyses are within the manuscript and its Supporting Information files.

**Funding:** This study/project is funded by the National Institute for Health and Care research (NIHR) Programme Grants for Applied Research, NIHR2000856 (https://fundingawards.nihr.ac.uk/award/NIHR200856) (awarded to author, C. S. E.). The views expressed are those of the authors and not necessarily those of the NIHR or the Department of Health and Social Care. The funders had no role in study design, data collection and analysis, decision to publish, or preparation of the manuscript

**Competing interests:** The authors have declared that no competing interests exist.

times. Facilitators included clarity about credibility and quality; inclusive content; and in-person assistance. Recommendations included: Education and Persuasion e.g., online and offline promotion and endorsement by healthcare professionals and peers; Training and Modelling e.g., accessible training to enhance searching skills and credibility appraisal; and Environmental Restructuring and Enablement e.g., modifications to ensure online information and support are simple and easy to use, including video/audio options for content.

## Conclusions

Given that access to many sexual health services is now digital, our analyses produced recommendations pivotal to increasing access to wider sexual healthcare among underserved populations. Implementing these recommendations could reduce inequalities associated with accessing and using online sexual health service.

## Introduction

Over the past decade, the online delivery of sexual healthcare has increased, accelerated by the COVID-19 pandemic [1–4]. Such healthcare includes online postal self-sampling (OPSS) for sexually transmitted infection (STI) and blood borne virus (BBV) testing [e.g. 5–8]. More complex online clinical care pathways are also in development, such as the eSexual Health Clinic for accessing STI test results and treatment [9] and ePrEP for accessing HIV prevention medication, pre-exposure prophylaxis (PrEP) [10]. For many people, the initial steps to accessing sexual healthcare, both online and traditional (i.e., in-person/phone), are seeking sexual health information and support online [11–15]. Our definition of seeking online sexual health information is inclusive, referring to searching for, finding, understanding, and applying information from the internet [e.g. 14–16] typically found through search engines. Equally, regarding seeking online sexual health support, we refer to finding and using text-based interactions for answers to a range of sexual health queries. These include tools such as live chats and chatbots (synchronous communication with a trained professional (live chats) or with automatic responses (chatbots)) and email or short-messaging service (SMS) text exchange (asynchronous communication with a trained professional [e.g. 17–22]). See S1 Table for a list of examples. Together, these two steps (seeking online sexual health information and support) form a digital doorway to wider sexual healthcare [23–27].

Online sexual healthcare can overcome common barriers to accessing traditional sexual health services, offering privacy and convenience [e.g. 12, 28, 29]. However, many may struggle to access and use online sexual healthcare due to inequalities patterned by socio-economic demographics, such as gender, sexual identity, ethnicity, and socio-economic status [3, 30–40] (i.e., underserved populations [41]). Further, for people to engage with and pass through the digital doorway to wider sexual healthcare, they require sufficient digital literacy (skills to perform tasks and solve problems in digital environments [42]) and health literacy (capability to understand, evaluate, and use information and services to make choices about health [43]). This complex intersection of socio-economic factors precluding access to healthcare for those who often bear a disproportionate burden of STIs [e.g. 44] illustrates how the provision of online sexual healthcare has the potential to widen inequalities amongst underserved populations.

To prevent widening inequalities in access to online sexual healthcare, it is vital to understand the barriers and facilitators to the digital doorway among underserved populations,

theorise the factors that underpin barriers and facilitators, and then identify appropriate theoretically informed recommendations for change [45]. While some research regarding barriers and facilitators has been conducted, the existing literature base is outdated [46–52] or uses exclusively quantitative methods [53–59]. Thus, there is an absence of contemporary, in-depth research. Moreover, to our knowledge, there are no studies identifying evidence-based and theoretically informed recommendations for seeking online sexual health information and support among underserved populations. Therefore, using a behavioural science approach, we first aimed to identify barriers and facilitators to two key elements of the digital doorway: i) seeking online sexual health information and ii) seeking online sexual health support. Subsequently, we aimed to propose theory-informed recommendations to improve these two access points. We developed three research questions (RQs): RQ1) What are the barriers and facilitators to seeking online sexual health information among underserved populations?; RQ2) What are the barriers and facilitators to seeking online sexual health support among underserved populations?; and RQ3) What evidence-based and theoretically informed recommendations can be made to enhance seeking online sexual health information and support among underserved populations?

## Methods

### Design

A behaviourally focused cross-sectional exploratory qualitative approach, conducted as part of the SEQUENCE Digital Programme (https://www.sequencedigital.org.uk/).

### Applying a behavioural lens

High quality applied behavioural science requires a considered understanding of the specific behaviour(s) that are intended to be changed by an intervention [45, 60]. Within the broad behavioural system of 'accessing and using online sexual healthcare' we identified seven distinct yet interconnected behavioural domains (see S1 Fig). Here, we focus on the first of these two domains that we consider to constitute the digital doorway: 1) seeking online sexual health information and 2) seeking online sexual health support.

### Participants

Inclusion criteria were: 1) never ordered/used or struggled to order/use an STI self-sampling kit (to recruit participants of lower digital literacy); 2) aged 16+; 3) sexually active; 4) had phone and internet access to enable data collection; 5) lived in the UK; and 6) spoke English well enough to participate in an interview. Further, using PROGRESSPlus (PROGRESS: Place of Residence, Race/Ethnicity, Occupation, Gender/Sex, Religion, Education, Socio-economic Status, Social Network; Plus: e.g., Age, Sexual Orientation, Disability) [61, 62], we developed a sampling framework (see S2 Table) to purposefully recruit participants from a range of underserved populations. In line with Braun and Clarke [63], we did not seek to meet data saturation. Instead, prior to recruitment, a sample of 35 was agreed as appropriate to meet the sample targets and sufficient Information Power [64]. After 35 interviews, we reviewed the data and were satisfied that Information Power has been attained. See Table 1 for the characteristics and demographics of the final sample.

### Recruitment

Representatives from five regional National Health Service (NHS) Trusts/Boards (i.e., organisational areas), two non-governmental organisations (NGOs), and one community college

**Table 1. Participant self-reported socio-economic demographic characteristics.**

| | Variable | n | % [a] |
|---|---|---|---|
| **Age** | 16–24 | 8 | 22.9 |
| | 25–34 | 10 | 28.6 |
| | 35–44 | 13 | 37.1 |
| | 45–54 | 1 | 2.9 |
| | 55–64 | 1 | 2.9 |
| | 65+ | 1 | 2.9 |
| **Ethnicity** [b] | Asian, Asian British or Asian Welsh: Chinese | 1 | 2.9 |
| | Asian, Asian British or Asian Welsh: Pakistani | 4 | 11.4 |
| | Black, Black British, Black Welsh, Caribbean or African: African | 3 | 8.6 |
| | Mixed or Multiple ethnic groups: Other Mixed or Multiple ethnic groups | 1 | 2.9 |
| | White: Irish | 1 | 2.9 |
| | White: English, Welsh, Scottish, Northern Irish, or British | 19 | 54.3 |
| | Other self-identified groups (e.g., Hungarian, Italian, Jewish) | 5 | 14.3 |
| **Sexual orientation** | Bisexual/queer women | 6 | 17.1 |
| | Bi/pansexual (gender diverse) | 2 | 5.7 |
| | Gay/bisexual men | 9 | 25.7 |
| | Heterosexual/straight men | 6 | 17.1 |
| | Heterosexual/straight women | 10 | 28.6 |
| | No response | 1 | 2.9 |
| **Gender** | Cisgender men | 16 | 45.7 |
| | Cisgender women | 16 | 45.7 |
| | Gender diverse (non-binary, trans masc) | 2 | 5.7 |
| **Education** | Secondary/High school | 6 | 17.1 |
| | College (introductory/foundational vocational award to diploma) | 12 | 34.3 |
| | University (Undergraduate student) | 5 | 14.3 |
| | University (Undergraduate degree achieved) | 5 | 14.3 |
| | University (Postgraduate degree achieved) | 5 | 14.3 |
| | No response | 1 | 2.9 |
| **Occupation** | Unemployed | 11 | 31.4 |
| | Student (full-time/part-time) | 9 | 25.7 |
| | Employed (full-time/part-time) | 13 | 37.1 |
| | Retired | 1 | 2.9 |
| **Area of deprivation** [c] | IMD Quartile 1 (most deprived) | 9 | 25.7 |
| | IMD Quartile 2 | 8 | 22.9 |
| | IMD Quartile 3 | 6 | 17.1 |
| | IMD Quartile 4 | 2 | 5.7 |
| | IMD Quartile 5 (least deprived) | 6 | 17.1 |
| | No postcode (i.e., homeless) | 1 | 2.9 |
| | No response | 2 | 5.7 |
| **Difficulty making ends meet** [d] | Yes | 9 | 25.7 |
| | Sometimes | 3 | 8.6 |
| | Not anymore but have in the past | 2 | 5.7 |
| | No | 20 | 57.1 |

(*Continued*)

**Table 1.** (Continued)

| | Variable | n | % [a] |
|---|---|---|---|
| **Belong to any particular religion or faith** | No (never, not anymore) | 20 | 57.1 |
| | Maybe (e.g., spiritual, agnostic) | 2 | 5.7 |
| | Yes | 12 | 34.3 |
| **Religion or faith** | Christianity | 7 | 20.0 |
| | Islam | 4 | 11.4 |
| | Judaism | 1 | 2.9 |
| **Disability** | Learning difficulty (n = 34) | 10 | 28.6 |
| | Mental or physical disability (n = 33) | 17 | 48.6 |
| **Disability reduces ability to carry out day-to-day activities** | Reduces ability to carry out day-to-day activities | 12 | 34.3 |
| | Sometimes reduces ability to carry out day-to-day activities | 1 | 2.9 |
| | Does not reduce ability to carry out day-to-day activities | 3 | 8.6 |
| **First language** | English | 26 | 74.3 |
| | Non-English (Italian, Indonesian, French, Hungarian, Yoruba, Urdu, Gaelic) | 8 | 22.9 |
| **Country born** | England | 13 | 37.1 |
| | Scotland | 13 | 37.1 |
| | Non-UK (Italy, Indonesia, Switzerland, Hungary, Nigeria) | 8 | 22.9 |

[a]Participant demographics for one participant were not obtained, table includes demographics for n = 34, except where participants did not wish to answer the question. Percentages are calculated for N = 35.
[b]Reported using the Official National Statistics classifications [68]
[c]Index of Multiple Deprivation [69]
[d]"Making ends meet" = ability to pay for essentials needed to live.

across England and Scotland referred interested potential participants to the research team. One NGO served people with disabilities and learning difficulties, the other, people who identify as LGBTQI+ and Muslim. The community college served people with low educational attainment living in a deprived area. We then telephoned each referred participant to verify eligibility according to our inclusion criteria, screen them against the target sampling frame, collect demographics and information on internet use (see S3 Table), and schedule an interview (phone, video, or face-to-face). The sample was monitored closely throughout recruitment by checking the demographics of potential participants against the existing sample and target sampling frame. As sample targets were met, we selectively recruited only people of characteristics and demographics for which targets had not yet been met and liaised with NHS and NGO representatives about targeting relevant potential participants.

## Materials

We composed a questionnaire-based assessment of eligibility to take part in the study, demographics (based on the PROGRESSPlus framework [61, 62]), and experience and skills using technology and the internet (S3 Table). In line with inclusive guidance [65, 66], the questions asked regarding socio-economic demographics, such as gender identity and ethnicity, were open questions to capture how participants self-identified.

We also developed an interview topic guide (S4 Table) and supporting visual aids (e.g., S2 Fig) to explore participants' barriers (e.g., what makes it difficult, what are the drawbacks) and facilitators (e.g., what makes it easy, what are the benefits) to both seeking online sexual health

information and support. Example topic guide questions were "What has made it difficult for you to search for sexual health information online?" and "What would make it easy for you to use an email or text service to get sexual health support online?". Alongside the topic guide questions, we asked follow up questions to participants' responses, for example, "You said that a barrier to searching for sexual health information online was privacy, can you tell me more about that?". Barriers and facilitators could be either experiential (i.e., actually experienced) or hypothetical. The visual aids depicted the two behavioural domains of seeking online sexual health information and support. Respectively, these visual aids showed search engines with the typed words "sexually transmitted infection (STI) symptoms" for 'seeking online sexual health information' and examples of a sexual health live chat and SMS text exchange with a healthcare provider (HCP) for 'seeking online sexual health support'.

## Participant and patient involvement and engagement

For material development, we consulted public and patient involvement and engagement (PPIE) representatives (N = 12) of diverse ages, genders, ethnicities, sexual orientations, religions, and experiences of disability, learning difficulties, and digital STI healthcare. The representatives offered intersectional perspectives and advice on our questionnaire-based assessment of participant demographics and internet use (n = 5) and the interview topic guide and visual aids (n = 7) to be used within data collection. Consent to share PPIE representatives' demographic information was not obtained.

## Procedure

One-to-one participant-led semi-structured interviews (duration range 38–82 minutes, $M$ = 60 minutes) were conducted remotely (video call n = 7, phone n = 23, or face-to-face n = 5, all by JMcL, between 15.10.21 and 18.03.22). Prior to the interview, we provided participants with an information sheet, consent form, and the visual aids by email or WhatsApp for remote interviews and hard copy for face-to-face interviews. At the beginning of the interview, for remote interviews, participants provided verbal informed consent, recorded (by JMcL) using an encrypted recorder, and for face-to-face interviews, participants provided written informed consent At the end of the interviews, all participants were offered a shopping voucher (value £30) and were provided with a list of sexual health resources (S5 Table) either by email, WhatsApp or hard copy.

## Analysis

For RQs one and two, using NVivo (version 12), we conducted inductive thematic analysis at semantic level to identify barrier and facilitator themes for each of the behaviours, following Braun and Clarke's [63] five steps (see Fig 1). First we familiarised ourselves with the data (step 1) then we systematically described data using brief summary barrier or facilitator statements (step 2). We then grouped similar summary statements to identify barrier and facilitator sub-themes (step 3) and grouped similar sub-themes to generate higher level barrier and facilitator themes, gaining insights into patterns across participant sub-groups (step 4). Finally, we reviewed the themes to ensure they matched the original quotes and finalised their names (step 5). Each of the steps were initially conducted by an experienced health psychology researcher (JMcL) then audited by an expert behavioural scientist (PF). Disagreements on the describing, grouping, and naming of themes were resolved through discussion until consensus was reached. An inter-disciplinary team of clinicians and clinical researchers, human computer interaction specialists, sociology researchers, and public health experts also had oversight of the analysis. Overall, the themes provide a rich description of what was learned from

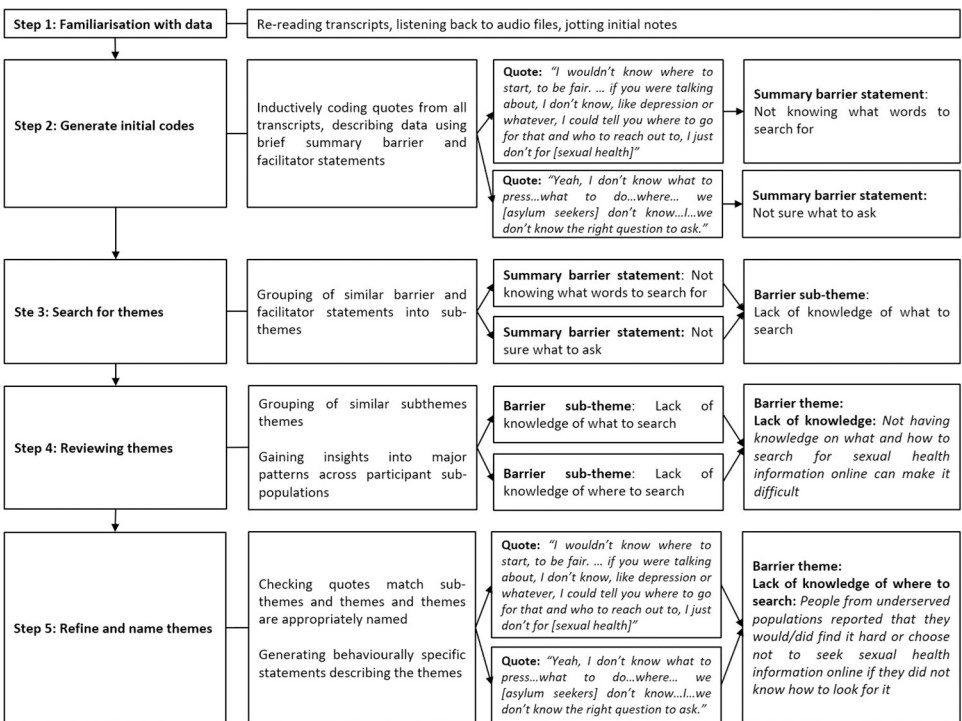

**Fig 1. The thematic analysis process, following Braun and Clarke's [63] five steps, using examples from data for seeking online sexual health information.**

analysing the whole dataset, reflecting all participants. Insights into the sub-population patterning of barriers and facilitators has also been provided; however no formal analysis across participants groups was conducted for this (e.g., approaches such as framework analysis [67]).

In line with Braun and Clark [63], thematic analysis was selected for the analysis, as it offered identification of patterns of important issues to participants, enabling us to determine the key barriers and facilitators to seeking online sexual health information and support across diverse underserved populations. Additionally, inductive analysis (i.e., themes are derived bottom up from the data without trying to fit it into a pre-existing coding frame or the researcher's analytic preconceptions) was chosen, as the goal of the research was to specify participant-led barriers and facilitators. Neither the topic guide nor the analysis were approached with pre-conceived notions of the data or results. Further, semantic level analysis (i.e., themes are identified from the explicit or meanings of the data, not anything beyond what a participant has said) was selected as this offers an objective description of the data, rather than subjective interpretation of underlying assumptions or deeper meanings within the data.

For RQ three, using the BCW approach [45], barrier and facilitator themes were mapped onto appropriate components of the COM-B Model [68] (by JMcL and checked by PF). The COM-B model posits that behaviour is determined by '**Capability**' (physical and psychological attributes of a person), '**Opportunity'** (physical and social attributes of a person's environment), and/or '**Motivation'** (a person's reflective and automatic mental processes). The COM-B components were then matched to relevant Intervention Functions (nine broad categories of potential interventions to change the capability, opportunity and/or motivation to engage in a behaviour) [45] (by PF, audited by JMcL and JMacD). Subsequently, we drew on our collective expertise and the Behaviour Change Technique Taxonomy version 1 (BCTTv1) [69] to operationalise the Intervention Functions into recommendations (conducted by PF

and JM, audited by JMacD). The recommendations were reviewed by an interdisciplinary team including sexual health clinicians, public health researchers, and human computer interaction specialists. All BCW analyses were conducted by health psychology researchers who have completed the BCTTv1 training (https://www.bct-taxonomy.com/). Discrepancies in BCW coding were resolved through discussion between JMcL, PF and, latterly, JMacD.

### Ethics

Written ethical approval for this study was granted by the East of England—Cambridge South Research Ethics Committee (REC) (reference 21/EE/0148) and Glasgow Caledonian University REC (reference HLS/NCH/20/045).

## Results

### Participants

Participants (N = 35) (see Table 1) ranged in age from 18–70 (*M* = 34 years) and were diverse, representing several underserved populations: 51% (n = 18) lived in the most deprived areas of the UK (as defined by the Index of Multiple Deprivation [70]); 51% (n = 18) had no higher (i.e., university) education; 40% (n = 14) were of a minoritised ethnic group, of which, five were from an ethnic group other than White; 23% (n = 8) did not speak English as their first language; 49% (n = 17) had a mental or physical illness or condition lasting 12 months or more; and 29% (n = 10) had a learning difficulty. The majority of participants reported owning a digital device (e.g., mobile phone or laptop) to access the internet (n = 30, 86%) and using the internet every day (n = 29, 83%) for a wide range of activities, most frequently, social media (n = 22, 63%), work including research (n = 17, 48%), streaming TV shows or videos (n = 10, 29%), searching the internet for information (n = 9, 26%), and checking news (n = 5, 14%) and emails (n = 5, 14%). Over half of the participants described themselves as having 'high' level skills using the internet (i.e., digital literacy) (n = 20, 57%), 12 reported 'medium' (34%), and 2 reported 'low' (6%). Over a third (n = 12, 34.3%) had never searched for any sexual health information online, none had ever used a live chat or email or text exchange service for getting sexual health support, and few had used other online sexual health services such as booking an appointment (n = 4, 11.4%) or ordering medication through a private (non-state-funded) clinic (n = 1, 2.9%). Moreover, the majority had never ordered a postal STI self-sampling kit (n = 24, 68.6%) and a few reported having tried and struggled to order (n = 3, 9%), or use (n = 7, 20%), a self-sampling kit for STIs and blood borne viruses. See S6 Table for an overview of participant data regarding experience and self-rated skills of using the internet.

### RQ1) What are the barriers and facilitators to seeking sexual health information online among underserved populations?

Table 2 details the nine barrier and eleven facilitator themes to seeking online sexual health information, with indicative data extracts, and their corresponding COM-B components.

   **Barriers to seeking online sexual health information.**   Barrier themes relating to **'Capability'** were '*Lack of awareness'; 'Lack of familiarity'; 'Lack of knowledge of where to search';* and '*Communication and understanding difficulties'*. This cluster of barriers related to a lack of important knowledge about the existence of sexual health information and how to access and understand it. Regarding insights into sub-population patterning of barriers, it appeared that the barrier '*Lack of awareness'* was noted despite self-reported high skills using the internet or amount of time spent on the internet. Additionally, '*Lack of familiarity'* appeared to be of particular issue for those who self-reported as having lower digital literacy (e.g., had a self-

**Table 2. Barrier and facilitator themes and corresponding COM-B components for seeking online sexual health information among underserved populations.**

| Theme and theme summary | Illustrative quote | COM-B component |
|---|---|---|
| **Barriers to seeking online sexual health information** | | |
| **Lack of awareness**<br><br>*People from underserved populations reported that they found it hard to seek sexual health information online. . . if they lacked awareness that there is sexual health information online* | *"being closeted, I struggled to know where to go, who to speak to about it [sexual health] and if I'm honest with you, I have had no experience of actually sexual health online whatsoever. [. . .] I think some of the issues is just I think I didn't really know that you could actually search for sexual health related support. I thought it was always you'd have to visit a professional in person or go through my GP for referral, so that's why I've never really thought about searching"* **(M, 40, high self-reported digital literacy, never ordered an OPSS [a] kit; sexual, ethnic, and religious minority)** | **'Capability'**<br><br>Physical and psychological attributes of a person e.g., mental functioning and resources |
| **Lack of familiarity**<br><br>*People from underserved populations reported that they would/ did find it hard or choose not to seek sexual health information online. . . if they lacked familiarity with using the internet or digital devices to search for information and were used to and comfortable with phone and in-person services* | *"I've no(t) actually tried it [. . .]. I'll probably start Googling stuff to do with sexual health, since this is brought to my attention. I've never really thought about it, I've always just, is what I say is, I just booked up for an appointment to get sexual health."* **(M, 61, self-reported medium digital literacy, never ordered an OPSS kit; sexual minority, unemployed, no higher education, living in deprived area)** | |
| **Lack of knowledge of where to search**<br><br>*People from underserved populations reported that they would/ did find it hard or choose not to seek sexual health information online. . . if they did not know how to begin looking for it* | *"I wouldn't know where to start, to be fair. Like I don't. . .like I said, there's no part. . .there's nothing really being promoted of. . .do you know what I mean? Cause I think what like, I wouldn't know where to start. Whereas, if you were talking about, I don't know, like depression or whatever, I could tell you where to go for that and who to reach out to, I just don't for [sexual health], you know?"* **(F, 28, self-reported high digital literacy, never ordered an OPSS kit; sexual, religious, and ethnic minority)** | |
| **Communication and understanding difficulties**<br><br>*People from underserved populations reported that they would/ did find it hard or choose not to seek sexual health information online. . . if they had difficulty reading, spelling, or there is medical jargon* | *"Sometimes I find it a bit too. . . some of it I find it a bit hard to look up on the internet because sometimes I can spell some of it but sometimes it can be hard for me."* **(M, 30, self-reported low digital literacy, never ordered an OPSS kit; learning difficulty, no higher education, unemployed, living in deprived area)** | |
| **Volume of information sources**<br><br>*People from underserved populations reported that they would/ did find it hard or choose not to seek sexual health information online. . . because of the overwhelming volume of sources returned in a search engine* | *"there's so much options that you're like which one? You understand? And I think the [real] one supposed to be at the top page. You understand what I mean? The one you're looking for should be on the top page so that people not get confused. For me I get confused so many times. [. . .] And I think even NHS (sites), [. . .] there's so much comes before the actual one. So, you get mixed up. You don't even know which one is which. So, I think that's the only thing that can make it difficult. [. . .] For me, I get confused."* **(F, 40+, self-reported high digital literacy, never ordered an OPSS kit; ethnic minority, English not first language)** | **'Opportunity'**<br><br>Physical and social attributes of a person's environment e.g., inanimate and interpersonal resources |
| **Lack of and inclusivity and personal relevance**<br><br>*People from underserved populations reported that they would/ did find it hard or choose not to seek sexual health information online. . . when resources were heteronormative, cisnormative, or perceived to lack personal relevance e.g., regarding gender differences in STI symptoms* | *"Yeah, well, now some people in some sexual health services are changing towards 'people with vaginas', 'people with penises', 'assigned male at birth', 'assigned female at birth'. But often it's like 'women' and 'men', which is, like obviously women are people with vaginas, men are people with penises, but that can be really kind of. . .not upsetting but it can make you uncomfortable if you're not a person who identifies that way."* **(NB trans masculine, 21, self-reported medium digital literacy, struggled using an OPSS kit; sexual minority, unemployed, living in deprived area)** | |

*(Continued)*

**Table 2.** (Continued)

| Theme and theme summary | Illustrative quote | COM-B component |
|---|---|---|
| **Concerns about privacy**<br><br>*People from underserved populations reported that they would/did find it hard or choose not to seek sexual health information online. . . if they were concerned about others witnessing their search or accessing their search history* | *"Maybe if you have a shared computer and you don't want the history on there. [. . .] Some people might want to keep their searching private and not want anyone else to know what they're looking at. [. . .] You might have a partner who you don't want to know that you have slept with someone else." (F, 24, self-reported high digital literacy, never ordered an OPSS kit; living in deprived area)* | **'Motivation'**<br><br>Reflective and automatic mental processes e.g., conscious thought processes |
| **Concerns about credibility**<br><br>*People from underserved populations reported that they would/did find it hard or choose not to seek sexual health information online. . . because they were uncertain about the trustworthiness of sites and concerned about misinformation* | *"[. . .] if I'm not feeling well, I tend not to. . .I try not to look at the internet for answers anymore. Unless I. . .like if I knew it was an established website then that would be different. [. . .] Like a. . .like a. . .NHS first or like a registered charity, that they would. . .that deals with specific problems. I don't. . .I don't fancy just like searching in things onto google or whatever search drive and taking their answers for granted. It would have to be a recognised body or institution that I would be like dealing with." (M, 43, self-reported medium digital literacy, never ordered an OPSS kit; learning difficulty, no higher education, living in deprived area)* | |
| **Concerns about adequacy/inferiority (compared to in-person/phone)**<br><br>*People from underserved populations reported that they would/did find it hard or choose not to seek sexual health information online. . . if they believed it would not be sufficient in meeting their needs and they would need to speak to a healthcare professional* | *"Yeah. Well, any symptoms, not all necessarily STIs, but health in general, I feel like the doctors would tell you not to look on the internet, because of how scary it can look, and how unprofessional the diagnoses are. So yeah, I guess, it wouldn't be my first reflex, to check on the on the internet, if I had something going wrong." (F, 22, self-reported medium digital literacy, never ordered an OPSS kit; sexual minority, English not first language)* | |
| Facilitators to seeking online sexual health information | | |
| **Knowledge of *how* and *for what* to search**<br><br>*People from underserved populations reported that they would/did find it easy or choose to seek online sexual health information. . . if they were familiar with using key words and formulating questions to locate more information* | *"Just I click on the wrong thing and then I click on something else and then I'm led to somewhere and I haven't even got a clue how I got there. What I've started to do is, like just say it was a sexually transmitted, I've just started to look up that, and 'sexually transmitted', I'd just look for that. But if it was the symptoms, I'd just put like the symptoms." (F, 43, self-reported medium digital literacy, never ordered an OPSS [a] kit; learning difficulty, living in a deprived area, unemployed)* | **'Capability'**<br><br>Physical and psychological attributes of a person e.g., mental functioning and resources |
| **Ability to determine credibility**<br><br>*People from underserved populations reported that they would/did find it easy or choose to seek sexual health information online. . . if they could distinguish between credible (e.g., governmental websites) and "sensationalist" (e.g., news websites) sources of information* | *"when it comes to health, when you do web searches of these things, or anything like that, a lot of people perhaps could go to the worst case scenario, perhaps, or they could go to a source that could be quite sensationalist. For example, sources like, the Express, isn't something that I would consider somewhere you would get health information from. However, sources like that remain a popular way to get information in regards to health, when it really shouldn't be the place many people should go to for that type of thing." (M, 25, self-reported high digital literacy, struggled using an OPSS kit; living in a deprived area)* | |

*(Continued)*

**Table 2.** (*Continued*)

| Theme and theme summary | Illustrative quote | COM-B component |
|---|---|---|
| **Credible sources being easy to find**<br><br>*People from underserved populations reported that they would/did find it easy or choose to seek sexual health information online. . . if credible sources were easy to find i.e., at the top of a search engine* | *"the information was just there, it wasn't like any links to click through, it was literally just in black and white right there [. . .] Because, well, the is NHS the first one that pops up when you search anyway, well it was for me anyway, so it was just right there."* **(F, 21, self-reported high digital literacy, never ordered an OPSS kit; ethnic minority)** | **'Opportunity'**<br><br>Physical and social attributes of a person's environment e.g., inanimate and interpersonal and resources |
| **Video/Audio options (alternative to text)**<br><br>*People from underserved populations reported that they would find it easy or choose to search for sexual information online. . . if sites had text to speech, audio options, or videos, to reduce the amount of text and reading* | *I just want to clarify here, if you were to search for the information by yourself, so there's not someone reading the information out for you, would it be easier for you if there were videos explaining the information, rather than reading the information, does that make sense?* **(Interviewer)** *Yeah.* **(M, 30, self-reported low digital literacy, never ordered a postal STI self-sampling kit; learning difficulty, no higher education, unemployed, living in deprived area)** | |
| **Simplicity of information and layout**<br><br>*People from underserved populations reported that they would/did find it easy or choose to seek sexual health information online. . . when sites used clear, easy to understand language, avoiding jargon and simple structures such as overview tabs* | *I suppose when you do click on the NHS website it is quite straightforward in terms of it has got the tabs at the top so it has an overview and symptoms, treatments and it usually always tells you where to refer to with more information, so, it's quite easy to understand I always find and it doesn't talk in too complicated medical terms that not everyone is going to understand.* **(F, 28, self-reported high digital literacy; struggled using an OPSS kit)** | |
| **Inclusivity and personal relevancy**<br><br>*People from underserved populations reported that they would/did find it easy or choose to seek sexual health information online. . . when sites were inclusive of and personally relevant to the reader, e.g., regarding gender, sexual identity, religion, drug use* | *"I get my information, when it comes to queer sexual health, from social media. It tends to be pages that are dedicated to that kind of thing or websites that are dedicated to queer orientated sexual health, because although NHS does have some information, it tends to be more difficult to get the information you need for using protection and that kind of thing [. . . . . .], I was really I about it because obviously you don't get taught about it at school and I think the NHS website lacks a lot with that kind of thing. So, yes, that's just the kind of things that I use them to find out about."* **(NB trans masculine, 21, self-reported medium digital literacy, struggled using an OPSS kit; sexual minority, unemployed, living in deprived area)** | |
| **Advertisements to raise awareness**<br><br>*People from underserved populations reported that they would find it easy or choose to seek online sexual health information online. . . if there was signposting both offline (e.g., community hubs, GPs) and online (e.g., social media, NHS websites)* | *"[What would make it easier to search for sexual health information online is] More information as to the direct type of place that we should be looking for. {. . .} So, it should be advertised a bit more, in that sense. If you need to do it [get sexual health information online], you can do it online [. . .] if [a community hub has] got it, there's. . .or even your own sexual health [local clinic] or whatever, there is all these places and numbers that you can get sexual health help but there is never a website saying, you can get it on this and you can do it online. There's nothing. There's no advertising saying that so, we [young vulnerable girls and women] don't know.* **(F, 29, self-reported medium digital literacy, never ordered an OPSS kit; sexual minority, learning difficulty, no higher education, unemployed, living in a deprived area)** | |
| **In-person assistance**<br><br>*People from underserved populations—particularly of lower digital literacy—reported that they would/did find it easy or choose to seek sexual health information online. . . if they had support, for example from family or a key worker, to search for, find, and understand relevant information* | *"What would make it [searching for sexual health information online] easier? Maybe have my daughter beside me, cause sometimes I know with Google, you type in a certain thing and about 20 things come up and you're clicking things that you think you've typed in but it's something totally different. So maybe having my daughter beside me or maybe one of the [support workers], one of them beside me would help me, make me do it more."* **(F, 43, self-reported medium digital literacy, never ordered an OPSS kit; learning difficulty, living in a deprived area, unemployed)** | |

*(Continued)*

**Table 2.** (Continued)

| Theme and theme summary | Illustrative quote | COM-B component |
|---|---|---|
| **Belief that information is credible** <br><br> *People from underserved populations reported that they would/ did find it easy or choose to seek sexual health information online… when sites were perceived as credible* | *"So, the NHS is, in general, regarded very favourably. […] As I understand it, the information there is all provided by someone who would be qualified to write on that subject. Obviously, being the institution that it is, is what makes it the source to go to, in my opinion. It's there, to me, it seems accountable with the information it provides, that's its role, it's not doing it for any other kind of interest, so to speak. It's there to provide the information that we need for the sake of, you know, for our health, basically, that's number one viewpoint that comes across to me." (M, 25, self-reported high digital literacy, struggled using an OPSS kit; living in a deprived area)* | **'Motivation'** <br><br> A person's reflective and automatic mental processes e.g., conscious thought processes |
| **Beliefs about equal/increased privacy (compared to in-person/phone)** <br><br> *People from underserved populations reported that they would/ did find it easy or choose to seek online sexual health information online… if they perceived it to be more private than phone or in-person or knew their privacy was safeguarded* | *"it is a lot more private. […] I think that is private, so, it's a lot easier but when you're sitting on the phone and people can hear you, people are in that conversation and people don't always want to…maybe we're told to give a bit of a description on the phone what a problem might be, we can't always do that because there are people there and taking it out the room doesn't mean anything because it's still…if people are as paranoid as what I can be, sometimes. People are always listening. So, aye, online's a lot more safer, private and a lot better." (F, 29, self-reported medium digital literacy, never ordered an OPSS kit; sexual minority, learning difficulty, no higher education, unemployed, living in a deprived area)* | |
| **Belief that online information is a good starting point** <br><br> *People from underserved populations reported that they would/ did find it easy or choose to seek online sexual health information… if they perceived it to be a good starting point for basic information (e.g., about STIs and symptoms), answering simple questions and reassurance (e.g., about STI prevention, symptoms and treatment)* | *"I feel like the internet is a good starting point. It wouldn't be like my final point, but good starting point. […] Because it's quite easy, it's easy obviously just do a quick google, have a little read, and then it wouldn't be my ending point, because I'd much rather go and actually speak to someone and like make sure that doubt [about symptoms], even though I know it's like with the information on the NHS website, would much rather hear it [information about sexual health online] from a person. […] Just because I wouldn't want to like, for me, myself, to like be misdiagnosed with anything or thinking…do you know what I mean, I wouldn't want to get confused myself, so I'd rather hear it from a professional." (F, 21, self-reported high digital literacy, never ordered an OPSS kit; ethnic minority)* | |

[a]OPSS = Online postal self-sampling kit. NHS = National Health Service, a state funded health service information site

perception of being, for example, a *"tech dinosaur"*). Further, *'Communication and understanding difficulties'* appeared to be of particular concern amongst those who reported having a learning difficulty or those whose first language was not English.

The barrier theme relating to **'Opportunity'** focussed on features of the digital environment: *'Volume of information sources'* and *'Lack of inclusivity and personal relevance'*. This barrier provides a sense of how some people find seeking online sexual health information overwhelming whilst navigating search engine results and finding relevant information. In relation to insights into sub-population patterning of barriers, it appeared that issues with a *'Lack of inclusivity and personal relevance'* was of particular concern for participants who identified as LGBTQIA+, were religious, or were an injecting drug user; for example, one such participant expressed the need for more information delineating the differences between STI symptoms and how the groin can be affected by injecting heroin.

Barrier themes relating to **'Motivation'** were negative perceptions of online sexual health information: '*Concerns about privacy*'; '*Concerns about credibility*'; '*Concerns about adequacy/ inferiority (compared to in-person/phone)*'. This cluster related to concerns about the internet as a source of sexual health information. In relation to insights into sub-population patterning of barriers, it appeared that '*Concerns about privacy*' was of particular issue for those who identified as LGBTQIA+ and/or religious; for example, Muslim LGBTQIA+ participants whose family did not know they were LGBTQIA+ reported concerns about their family seeing their search history.

**Facilitators to seeking online sexual health information.** Facilitator themes related to **'Capability'** centred around knowledge: '*Knowledge of how and for what to search*' and '*Ability to determine credibility*'. Regarding insights into sub-population patterning of facilitators, it appeared that '*Knowledge of how and for what to search*' was particularly important among participants who reported lower skills using the internet, noting this as a particular facilitator to finding relevant information. Additionally, '*Ability to determine credibility*' of information appeared to be particularly prominent among those who self-reported as having high skills using the internet.

Facilitator themes related to **'Opportunity'** focussed on features of the digital environment, promotion, and help: '*Credible information being easy to find*'; '*Video and audio options (alternative to text)*'; '*Simplicity of information and layout*'; '*Inclusivity and personal relevancy*'; '*Advertisements to raise awareness*'; and '*In-person assistance*'. In relation to insights into subpopulation patterning of facilitators, it appeared that '*Credible information being easy to find*' was particularly important for those who self-reported as having lower skills using the internet. Additionally, the '*Inclusivity and personal relevancy*' of information appeared to be particularly prominent among those who identified as LGBTQI+, religious, and injecting drug users. Similarly, '*Video and audio options*' were of particular importance among those who reported having a learning difficulty. Further, '*In-person assistance*' speaks to the need for help to seek online sexual health information, for example from a key worker or family member, among those who self-reported as having lower skills using the internet or with learning difficulties.

Facilitator themes relating to **'Motivation'** were positive perceptions of online sexual health information: '*Belief that information is credible*'; '*Beliefs about equal/increased privacy (compared to in-person/phone)*'; and '*Belief that online information is a good starting point*'. No subpopulation patterning of these facilitators were apparent.

**RQ2) What are the barriers and facilitators to seeking online sexual health support among underserved populations?.** Table 3 details eight barrier and eight facilitator themes

**Table 3. Barrier and facilitator themes and corresponding COM-B components for seeking online sexual health support among underserved populations.**

| Theme and theme summary | Indicative quotes | COM-B component |
|---|---|---|
| **Barriers to seeking online sexual health support** | | |
| **Lack of awareness** | *"If I'm being honest, I didn't even know they existed, the live chats, through the sexual health worker. [. . .] I'm one of the older generation, though I'm 61, so I'm kind of older and I actually just even got a smartphone and I'm not clued into all that. I've come across a few people that have not got any computer skills at all. At least I've got some." (M, 61, self-reported medium digital literacy, never ordered an OPSS [a] kit; sexual minority, unemployed, no higher education, living in deprived area)* | **'Capability'** |
| *People from underserved populations reported that they would find it hard to get sexual health support online. . . because they lacked awareness that online sexual health support exists* | | Physical and psychological attributes of a person e.g., mental functioning and resources |
| **Lack of familiarity** | *"I've used services through calling the services, and that's what I've always known. Because I've not used live chat services like that, I wouldn't know, so I wouldn't know what would be involved" (M, 25, self-reported high digital literacy, struggled using an OPSS; living in a deprived area)* | |
| *People from underserved populations reported that they would find it hard or choose not to get sexual health support online. . . if they lacked familiarity with using a live chat, email, or text exchange service and were used to phone or in-person services* | | |

*(Continued)*

**Table 3.** (Continued)

| Theme and theme summary | Indicative quotes | COM-B component |
|---|---|---|
| **Concerns about confidentiality and anonymity**<br><br>*People from underserved populations reported that they would find it hard or choose not to get sexual health support online. . . if they were concerned about the confidentiality of live chats and worried about others, such as partners or parents, finding out they may have STI or sexual health questions or concerns* | *"I mean as much as I think it's a good thing, I do. . .I would still have some concerns about putting in my full. . .you know, like my real name and stuff like that unless I really knew for sure I was dealing with a confidential company. . .or like the NHS"* **(M, 43, self-reported medium digital literacy, never ordered an OPSS kit; learning difficulty, no higher education, living in deprived area)** | **'Motivation'**<br><br>A person's reflective and automatic mental processes e.g., conscious thought processes) |
| **Concerns about the impersonal nature of online support**<br><br>*People from underserved populations reported that they would find it hard or choose not to get sexual health support online. . . if they perceived it to be impersonal and lacking capacity to provide reassurance* | *"Mhm. I guess, my first thought would be, how impersonal it can be. [. . .] I would say, it's more of a disadvantage, as in, yeah, it does feel a bit impersonal, and I feel, because of how personal sexual health is, that would be a bit. . .like a gap. . .that's weird for me."* **(F, 22, self-reported medium digital literacy, never ordered an OPSS; sexual minority, English not first language)** | |
| **Concerns about understanding and communication**<br><br>*People from underserved populations reported that they would find it hard or choose not to get sexual health support online. . . if they had difficulty with spelling or reading and/or were concerned about staff not being able to communicate clearly and effectively* | *"I'd probably be concerned that. . .whether I was able to understand what was being said in the first place."* **(M, 49, self-reported medium digital literacy, struggled ordering an OPSS kit; learning difficulty, no higher education, unemployed)** | |
| **Concerns about online support meeting their needs**<br><br>*People from underserved populations reported that they would find it hard or choose not to get sexual health support online. . . if they believed it would not be sufficient in meeting their needs and they would need to see a healthcare professional in-person* | *"Yeah, yeah, yeah, so as in like, if I was texting someone and I thought I had an STI, it would for me it would just be the same as Googling. I'd personally want to speak to someone physically to kind of possibly be reassured. You can't translate that on a message, you know?"* **(F, 28, self-reported high digital literacy, never ordered an OPSS; sexual, religious, and ethnic minority)** | |
| **Concerns about chatbots and automated responses**<br><br>*People from underserved populations reported that they would find it hard or choose not to get sexual health support online. . . if they knew or thought it was being delivered via automated responses instead of a qualified professional or trained member of staff due to concerns about the quality of the responses* | *"I don't really mind if it's trained staff or a doctor or a nurse, but I think I'd. . . not do it if it's a chatbot, because sometimes they can be not helpful. [. . .] Because. . . well, the chatbot I had experienced [for non-sexual-health-related help] only had like set responses and sometimes when you want to use the live chat you kind of have more complicated stuff to ask about, sometimes the set responses aren't that helpful, so it's more frustrating than just not doing it. [. . .] like I could just Google it instead of talking to a chatbot, cause like I said, if I'm going to do a live chat, it's probably something more complicated and it's like something I can't Google. [. . .] I'd use like a live chat or an email thing, if it's a healthcare professional on the other end."* **(M, 18, self-reported high digital literacy; never ordered an OPSS kit; sexual and ethnic minority, English not first language)** | |
| **Concerns about response wait times**<br><br>*People from underserved populations reported that they would find it hard or choose not to get sexual health support online. . . if there was a wait time for receiving response including hours to days via asynchronous communication or half an hour for synchronous communication* | *"I just probably wouldn't use that because it's just prolonging like whatever problem I need help with. [. . .] if I wanted to speak to someone and wanted to get help, then it doesn't really seem useful to be waiting for them a couple of days or a day or so for an answer, then send one back and then wait another day, it just doesn't really seem like. . .don't really see the point in it."* **(F, 21, self-reported high digital literacy, never ordered an OPSS kit; ethnic minority)** | |
| **Facilitators to seeking online sexual health support** | | |
| **Familiarity with online text-based support**<br><br>*People from underserved populations reported that they would find it easy or choose to get sexual health support online. . . if they had familiarity with using live chats, email, or text exchange services as modes of communication* | *"If you're very familiar with texting, and using your email, to me, it would be pretty convenient to use it, because obviously you're very familiar with that already."* **(M, 42, self-reported high digital literacy, struggled ordering an OPSS ᵃ kit; sexual and religious minority, living in deprived area)** | **'Capability'**<br><br>Physical and psychological attributes of a person e.g., mental functioning and resources) |

(*Continued*)

**Table 3.** (Continued)

| Theme and theme summary | Indicative quotes | COM-B component |
|---|---|---|
| **Immediate responses**<br><br>*People from underserved populations reported that they would find it easy or choose to get sexual health support online if responses were quick (i.e., immediate), as sexual health queries were typically perceived to be urgent* | *"I imagine it [sexual health live chat] would be quite good cos it would be like immediate…you're immediately directed to someone, so you can ask questions." (F, 21, self-reported high digital literacy, never ordered an OPSS kit; ethnic minority)* | **'Opportunity'**<br><br>Physical and social attributes of a person's environment e.g., inanimate and interpersonal and resources) |
| **Clear trustworthiness and high quality**<br><br>*People from underserved populations reported that they would find it easy or choose to get sexual health support online… if they knew the interaction was with a qualified/trained person and, for some, if it was a safe space for discussing LGBTQI+ issues* | *"If I knew this person was working for the NHS, I would… I immediately have a trust. I just trust them completely. I know sometimes there are breaches in confidentiality in all organisations, but I just immediately feel I can trust this person with very sensitive information. If it's a private company, I wouldn't feel the same [...] I wouldn't automatically trust them" (M, 70, self-reported medium digital literacy, struggled using an OPSS kit; sexual minority, living in deprived area)* | |
| **Personal feel**<br><br>*People from underserved populations reported that they would find it easy or choose to get sexual health support online… if it felt personal (e.g., had a friendly face, welcome invitation, and/or feels like a chat rather than formal)* | *"someone who kind of talks in their dialect so every time you come on to these chats it's a different speech pattern, it's a different stress pattern on words and you can see it there visually in front of you so you know it's someone different and you know it's a person, if that makes sense. [...] So, just having like a true voice, like allowing people to have that true voice, let the informality, let the slang be there… Maybe give someone, maybe like at the start when opting in for the chat, give someone the option like would you like a more formal chat or an informal chat or if there's another way to discuss that or ask that question." (NB, 27, self-reported medium digital literacy, never ordered an OPSS kit; sexual, ethnic, and religious minority, learning difficulty, living in deprived area)* | |
| **Advertisements to increase awareness**<br><br>*People from underserved populations reported that they would find it easy or choose to get sexual health support online… if there was promotion both offline (e.g., community hubs, GPs) and online (e.g., social media, NHS websites)* | *"Again, like I said, advertisement, more people talking about it [sexual health live chat services], like sexual health clinics and people opening up and talking about it but it's no…if we go to a sexual health team, we never hear anybody, you know you can go online and get this, we don't hear anything like that. It's just, right, see you next week" (F, 29, self-reported medium digital literacy, never ordered an OPSS kit; sexual minority, learning difficulty, no higher education, unemployed, living in a deprived area)* | |
| **In-person assistance**<br><br>*People from underserved populations reported that they would find it easy or choose to get sexual health support online… if they had help, e.g., from key workers, to find online services, type questions, and interpret responses* | *"Some people with disabilities who need…who get support would need to do it with support 'cause like me, probably wouldn't understand the long jargon words. And need help to spell" (M, 49, self-reported medium digital literacy, never ordered an OPSS kit; learning difficulty, no higher education, unemployed)* | |
| **Beliefs about equal/increased privacy (*compared to in-person/phone*)**<br><br>*People from underserved populations reported that they would find it easy or choose to get sexual health support online… if they perceived it to be more private than phone or in-person or knew their privacy was safeguarded* | *"it can feel a bit uncomfortable or a bit embarrassing going to a sexual health clinic in a way. So, it [a sexual health live chat/email/text service] would take that out of it as well. [...] I suppose it's because sexual health is still not something we just talk about openly, especially not as an Asian person. In the Asian community, we don't really talk about sex a lot, so I feel quite like I'm being watched when I do go into the clinic." (F, 43, self-reported high digital literacy, never ordered an OPSS kit; physical disability, no higher education, unemployed, living in a deprived area)* | **'Motivation'**<br><br>A person's reflective and automatic mental processes e.g., conscious thought processes) |
| **Beliefs about online support meeting their needs**<br><br>*People from underserved populations reported that they would find it easy or choose to get sexual health support online… if they perceived it to be sufficient to meet their needs, i.e., for general or specific queries, link to in-person services, to arrange a call back, help to order an OPSS, or to book an appointment* | *"as far as I understand, these live chat services [...] is to kind of get sexual health support and advice so, and what I understand by sexual health support could be, again, general information about kind of sexual health in general but also kind of support about, say, suppose I might just, I've just had sex and I'm concerned about this…information about a specific instance, for example, am I at risk, am I a risk? So, again, I don't think speaking to someone in person, for example, wouldn't necessarily be better than using the live chat. [...] so if it's just concerning a) some general questions about, say, what kind of services are available or b) general information about STIs and sexual health in general and c) information about…again, suppose I just had a sexual encounter that was deemed to be kind of a risk and I wanted to kind of speak to someone about it because I had some concerns then an online live chat would do absolutely fine." (M, 35, self-reported high digital literacy, struggled using an OPSS kit; ethnic minority, English not first language)* | |

[a]OPSS = Online postal self-sampling kit. NHS = National Health Service, a state funded health service information site

to seeking online sexual health support (via a synchronous live chat or asynchronous email or SMS text exchange), with illustrative extracts, and corresponding COM-B components.

**Barriers to seeking online sexual health support.**   Barrier themes relating to **'Capability'** were: *'Lack of awareness'* and *'Lack of familiarity'* with online text-based support. These themes revealed a lack of awareness of the existence of online sexual health support services and experience using digital technology and the internet for sexual health support. Regarding insights into sub-population patterning of barriers, it appeared that a '*Lack of familiarity'* was of particular issue for those who self-reported has having lower skills using the internet.

Barrier themes relating to **'Motivation'** were negative perceptions about digital sexual health support: '*Concerns about confidentiality and anonymity'; 'Concerns about the impersonal nature of online support'; 'Concerns about understanding and communication'; 'Concerns about online support meeting their needs'; 'Concerns about chatbots and automated responses';* and *'Concerns about response wait times'.* Participants consistently reported that they would not use online support for sexual health that had any wait times (i.e., was not instantaneous) to receive a response, or if they would be speaking to a 'chatbot' with pre-set or automated responses instead of a person, particularly a trained professional. In relation to insights into sub-population patterning of barriers, it appeared that '*Concerns about confidentiality and anonymity'* was of particular issue for those who identified as being LGBTQIA+, religious, or in a relationship; for example, those of a religious background noted concerns about parents or family members finding out they were sexually active. Additionally, *'Concerns about understanding and communication'* appeared to be particularly important among those who had a learning difficulty or whose first language was not English.

No barriers corresponded to **Opportunity**.

**Facilitators to seeking online sexual health support.**   One facilitator theme related to **'Capability'** was identified regarding knowledge: *'Familiarity with online text-based support services'.* In relation to insights into sub-population patterning of facilitators, it appeared that '*Familiarity with online text-based support services'* was of particular issue for those who self-reported as having lower skills using the internet.

Facilitator themes corresponding to **'Opportunity'** related to advertisements, help, and features of the digital environment: *'Immediate responses'; 'Clear quality and trustworthiness'; 'Personal feel'; 'Advertisements to increase awareness';* and' *In-person assistance'.* Regarding insights into sub-population patterning of barriers, *'In-person assistance'* illustrates the needs of participants who self-reported as having lower skills using the internet or those with a learning difficulty for help engaging and accessing online support.

Facilitator themes relating to **'Motivation'** were positive perceptions on online sexual health support: '*Beliefs about equal/increased privacy (compared to in-person/phone)'* and *'Beliefs about online support meeting their needs'.* No sub-population patterning of these facilitators were apparent.

**RQ3) What evidence-based and theoretically informed recommendations can be made to enhance seeking online sexual health information and support among underserved populations?.**   The BCW analysis identified potentially useful Intervention Functions to assist underserved populations in seeking online sexual health information and support: **'Education', 'Persuasion, 'Training', 'Modelling', 'Enablement',** and **'Environmental Restructuring'** (see S7 Table for definitions). From these Intervention Functions, we proposed a range of more specific recommendations outlined below. Tables 4 and 5 provide an overview of the BCW analysis of barriers and facilitators and proposed recommendations for online sexual health information and support, respectively.

**'Education' and 'Persuasion'.**   It may be critical to educate people from underserved populations, particularly those of lower digital literacy, that online sexual health information and

**Table 4. Intervention content from Behaviour Change Wheel to address barriers and facilitators to seeking sexual health information online.**

| Barriers | Facilitators | COM-B | Intervention Functions and recommendations for those providing sexual health information online |
|---|---|---|---|
| • Lack of awareness<br>• Lack of familiarity<br>• Communication and understanding difficulties<br>• Lack of knowledge of where to search | • Knowledge of *how* and *for what* to search<br>• Ability to determine credibility | Capability | **Education** and **Persuasion:**<br>1. Promote online sexual health information via social media, HCPs and NGOs with the professional endorsement of credible sites.<br>**Education**:<br>1. Provide information via interactions with service staff, leaflets or adverts detailing where to search (i.e., search engines/NHS website), what to search for (i.e., key terms), and how to identify credible sources (i.e., key logos).<br>**Training**:<br>1. Provide in-person and online training for actual or potential service users in using search engines and key terms, focus on developing skills to determine source credibility and understanding and applying information.<br>2. Provide contemporary guidance to assist with the phraseology of searches to make the most out of artificial intelligence.<br>3. Provide professional or peer-led opportunities for people to shadow trusted others<br>4. Provide online tutorials of how to use services.<br>5. Ensure all training is accessible to many (e.g., in-person as well as online provision, and audiovisual rather than text-based).<br>6. Explore partnerships with NGOs and other non-sexual health service providers, to identify where, and by who, this activity should be delivered.<br>**Modelling**:<br>1. Provide accessible examples, such as videos, or posters of how to search effectively, using peers that show diverse service users successfully searching for sexual health information online.<br>**Enablement**:<br>1. Provide an easily accessible, 'copy and paste-able' glossary of terms (i.e., 'gonorrhoea') that give simple access to the correct spelling for searches. |
| • Volume of information sources<br>• Lack of and inclusivity and personal relevance | • Credible sources being easy to find<br>• Inclusivity and personal relevancy<br>• Video/Audio options (alternative to text)<br>• Simplicity of information and layout<br>• Advertisements to raise awareness<br>• In-person assistance | Opportunity | **Environmental Restructuring**:<br>1. Advertise online information services both online and offline.<br>2. Ensure credible sources (i.e., NHS, registered charities) are accessible and simple to use and working with providers attempt to ensure that credible sources appear first in search results.<br>3. Ensure the use of LGBTQI+, religious, and drug user inclusive content, language and images throughout.<br>**Enablement:**<br>1. Where it is needed, provide in-person assistance to educate, train, or support people accessing and using online information services.<br>2. Provide inclusive sexual health information (language and visuals) for a range of communities (e.g., LGBTQI+ communities, people who use drugs).<br>3. Clearly differentiate sexual health information for different genders and anatomies.<br>4. Ensure that video and/or audio options are available as an alternative to reading text; avoid medical jargon and explain terms when they must be used. |
| • Concerns about privacy<br>• Concerns about credibility<br>• Concerns about adequacy/inferiority (compared to in-person/phone) | • Belief that information is credible<br>• Beliefs about equal/increased privacy (compared to in-person/phone)<br>• Belief that online information is a good starting point | Motivation | **Persuasion:**<br>1. Ensure that professionals, or peers, affirm the value and credibility of trusted sites within face-to-face interactions.<br>2. Ensure adverts can promote credible sites harnessing key opinion leaders with multiple followers e.g., social media influencers.<br>**Education** and **Persuasion:**<br>1. Provide information about the privacy and confidentiality of online information services; provide information about the likely adequacy of online information to meet people's needs (e.g. finding services vs seeking assessment of symptoms).<br>2. Provide transparent information about the limits of online information services clearly showing when in-person care should be sought. |

**Table 5. Intervention content from behaviour change wheel to address barriers and facilitators to seeking online sexual health support.**

| Barriers | Facilitators | COM-B | Intervention functions and suggested intervention content for those providing online sexual health support |
|---|---|---|---|
| • Lack of awareness<br>• Lack of familiarity | • Familiarity with text-based support services | Capability | **Education**:<br>1. Promote high-quality online sexual health support both online (i.e., social media) and offline (i.e., diverse professionals and/or peers and carers) and ensure that people are aware of what online support services is, and is not, good for (e.g. not for seeking diagnosis but for details about when to seek an in-person appointment).<br>**Training**:<br>1. Provide video tutorials of how to use online sexual health support, including top tips of what to do and not to do.<br>2. Provide posters with step-by-step instructions, and top tips.<br>3. Provide professional or peer-led opportunities for people to shadow trusted others<br>4. Provide online tutorials of how to use services.<br>**Modelling**:<br>1. Provide videos or posters, with examples of peers, to show service users successfully searching for and using online sexual health support. |
| | • Immediate responses<br>• Clear quality and trustworthiness<br>• Personal feel<br>• Advertisements to raise awareness<br>• In-person assistance | Opportunity | **Environmental Restructuring**:<br>1. Ensure text-based interactions are simple and easy to use as well as being synchronous (in real-time), provided by a trained professional with interpersonal skills.<br>2. Ensure the name and training/qualifications of the providers are stated.<br>3. Advertise online support services both online and offline.<br>**Enablement**:<br>1. Where it is needed, provide in-person assistance to educate, train, or support people accessing and using online support services. |
| • Concerns about confidentiality and anonymity<br>• Concerns about chatbots and automated responses<br>• Concerns about response wait times<br>• Concerns about online support meeting their needs<br>• Concerns about the impersonal nature of online support<br>• Concerns about understanding and communication | • Beliefs about equal/increased privacy (compared to in-person/phone)<br>• Beliefs about online support meeting their needs | Motivation | **Education** and **Persuasion**:<br>1. Provide information about the privacy and confidentiality of online support services.<br>2. Provide information about the likely adequacy of online support to meet people's needs (e.g. finding services vs seeking assessment of symptoms.<br>3. Provide transparent information about the limits of online support services clearly showing when in-person care should be sought.<br>**Education**:<br>1. Promote online sexual health support as simple and easy to use, personal, synchronous with immediate responses, and not reliant on automated responses<br>**Persuasion:**<br>1. Use professionals or peers to affirm value and credibility of trusted support services during face-to-face interactions<br>2. Advertise to promote credible support services harnessing key opinion leaders with multiple followers e.g., social media influencers<br>**Training**:<br>1. Train staff in managing situations where communication is genuinely challenging<br>**Enablement**:<br>1. Provide a useable, 'copy and paste'-able glossary of terms (i.e., 'gonorrhoea') for people to copy correct spelling |

support services exist, for example, through advertising online (e.g., via social media) and offline (e.g., via HCPs and NGOs), and persuade them to use these via endorsement by a range of credible professionals and peers. In particular, offline advertising will be crucial for reaching people of lower digital literacy. This education could be delivered via face-to-face interactions with service staff (e.g., HCPs and key workers), posters or leaflets, or posts on professional social media accounts (e.g., from HCPs or influencers) detailing where to search (i.e., search engines/NHS website), what to search for (i.e., key terms), and how to identify credible sources (i.e., key logos). It may also be important to inform people about the benefits and limits of online sexual health information and support and when in-person care should be sought i.e.,

that online information and support are appropriate for simple issues, such as symptom checking, finding clinics, booking appointments, and signposting, and not appropriate for complex healthcare, such as diagnosis. Further, education regarding the privacy and confidentiality of online sexual health information and support may be important, as well as promoting them as simple and easy to use, personal, synchronous with immediate responses, and not reliant on automated responses). It is critical that Education and Persuasion are delivered alongside Environmental Restructuring and Enablement to ensure online sexual health information and support are private/confidential and simple to use.

'Training' and 'Modelling'.   Training opportunities for necessary skill acquisition could be part of all services that offer online information and support, particularly for those who self-reported as having lower skills using the internet or with learning difficulties. Specifically, this could involve step-by-step instructions on using search engines, identifying and typing key terms, determining source credibility, and understanding and applying information. Training could be delivered in the form of videos, posters or leaflets, or professional or peer-led tutorials. To ensure accessibility in relation to digital provision, audiovisual training options should be offered alongside text-based and training should be provided both in-person as well as online. It might also be critical to provide modelling of peers successfully using online sexual health information and support, such as showing videos or images of peers from underserved populations searching for sexual health information such as STI symptoms or using a sexual health live chat to discuss safe sex. Further, online sexual health information and support service providers could explore partnerships with NGOs for delivery of training.

'Environmental Restructuring' and 'Enablement'.   Restructuring the online environment and enabling people from underserved populations to seek online sexual health information and support is essential. Particularly, it may be vital to ensure that online sexual health information and support services are clearly labelled as delivered by credible, trusted sources (i.e., NHS, registered charities). Further, such services should be designed to be simple to use, avoiding medical jargon and explaining terms when they must be used, particularly to support those who self-report as having lower skills using the internet and with learning difficulties. Additionally, enablement could involve providing a glossary of sexual health terms for examples of correct spelling and video and/or audio options for information. Online sexual information and support content could also be demonstrably tailored to a range of communities, such as gender, sexuality, and religious minoritised groups, and people who use injecting drugs. Further, in-person assistance could enable people of lower digital literacy and with learning difficulties to access and use sexual health information and support. Finally, online sexual health information and support service providers could ensure that any text-based interactions are synchronous (in real-time) and delivered by a trained professional, stating the name and training/qualifications of the professional involved.

## Discussion

This study is the first to detail barriers and facilitators to seeking online sexual health information and support amongst underserved populations and propose specific recommendations to enable underserved populations to find and use online sexual health information and support. These findings can be applied to existing and novel online sexual health information and support services to improve access to this digital doorway to wider sexual health services.

### Barriers and facilitators to passing through the digital doorway

Key barriers to people from underserved populations seeking online sexual health information and support were a lack of awareness of their availability and familiarity with using them;

privacy concerns; and the perception of as inadequate to meet varied and complex sexual health needs and inferior to traditional information and support services (e.g., in-person appointment or phone call with a HCP). For seeking online sexual health information, specifically, barriers were navigating the overwhelming volume of information and different sources of and the perceived lack of inclusivity and relevance of information, particularly on government websites. For seeking online sexual health support, specifically, barriers were chatbots and automated responses; asynchronous communication; and wait times for responses.

Important facilitators to seeking online sexual health information and support were the perceived benefit of increased privacy compared to traditional services; the provision of video and/or audio options as alternatives to text; the presentation of information and support in a simple way–such as step by step and without jargon; and in-person assistance, for example, from key workers, family members, NGO staff, or GPs. For seeking online sexual health information, specifically, facilitators were inclusivity and personal relevancy of sexual health information, particularly for people from LGBTQI+ populations, religious backgrounds, and injecting drug users. For seeking online sexual health support, specifically, facilitators were synchronous communication; the perception of online support as acceptable for simple tasks such as appointment booking, general sexual health advice, and signposting to services; and delivery by qualified personnel.

An important consideration here is that none of the participants had ever used an online sexual health support service, despite their availability [e.g. 17–22], thus these barriers and facilitators were exclusively hypothetical. Additionally, barriers and facilitators differed in intensity and nature among subgroups; for example, privacy and confidentiality concerns were expressed by most participants but for different reasons and thus would likely differ among those not represented in this study.

Some of the barriers and facilitators are novel to this study. To the authors' knowledge, low awareness of online sexual health information and support has not been reported previously and was common, even among participants who reported using the internet 'all the time'. Additionally, the perceived inferiority and inadequacy of online sexual health information compared to traditional, such as in-person or phone, sources of sexual health information was novel to this study. Further, the unanimous rejection of chatbots and automated responses for sexual health support, due to the perceived complexity of sexual health, was novel to this study. This is in contrast to previous research showing that chatbots for sexual health were overall perceived to be useful for ongoing care or anonymous sex education (by samples of largely white heterosexual adult women) [59, 71]. However, beliefs about the inferiority of online services may change, as generative artificial intelligence becomes more commonplace and trusted [72].

On the other hand, many of our barrier and facilitators findings are consistent with previous literature, offering a novel perspective from a diverse sample of underserved populations. Privacy benefits and concerns [38, 57, 71, 73–75] and the value of video and audio options and simplicity [e.g. 29, 38, 75] have been previously reported amongst a range of populations. Further, the overwhelming volume of sources and information, need for inclusivity and personal relevancy of information, and preferences for online support to be delivered synchronously have been reported previously among sexual minority women [76] and African American youth [77], and young people [38, 78, 79]. Moreover, concerns about the adequacy of online information and support has been reported previously [71, 75, 80] and the acceptability of online support for simple tasks is in line with research regarding HCPs' views of chatbots for sexual health [75]. Overall, the consistency of these findings with research with other populations indicates that addressing the barriers and enhancing the facilitators from this current

study will improve online sexual health information and support for many beyond this sample.

## Recommendations for opening the digital doorway to wider sexual healthcare

We identified a range of theory informed Intervention Functions to improve access to the digital doorway to wider sexual healthcare, including **'Education', 'Persuasion', 'Training', 'Modelling', 'Environmental Restructuring',** and **'Enablement'**. First, advertising and promotion of services that provide online sexual health information and support and HCP's endorsing online services and supporting patients to use them is vital (**'Education'** and **'Persuasion'**). Together, these findings resonate with wider research on telehealth *per se* [81], highlighting the importance of marketing and communication activities to increase awareness of online services to enable equitable access. Additionally, interventions could include online services offering audio-visual forms of communication, such as text-to-speech or cartoon animations. Moreover, our recommendations include the provision of inclusive and personally relevant information and support, such as information about sexual health relevant to those of diverse sexual and/or gender identities, religious backgrounds, and drug use (**'Environmental Restructuring'** and **'Enablement'**). This is in line with previous research showing that tailoring of information can positively influence acceptability of interventions and address barriers to care [82–84]. Further, interventions could include training HCP's on how to use and promote online sexual health services effectively, in line with previous research regarding reducing digital inequalities [81]. Further, professional endorsement of high-quality online services for sexual health information and support is also known to be important [85]. Such endorsements should stress the services' credibility, useability, and details of its functionality (e.g., trained professional will interact in real time). Interventions could also include embedding optional training for service users regarding how to use online services (**'Training'** and **'Modelling'**).

## Implications

Within intervention development, context is critical for the likely success of the intervention [45, 86, 87]. As such, while the findings of this study are solely derived from participants living in Scotland and England, we are confident that they should be transferable to countries with similar healthcare settings and digital infrastructure, such as other countries in the UK not represented in this sample (i.e., Northern Ireland and Wales) or Australia, Canada, and New Zealand. We are less confident about the transferability of the findings to low-and-middle-income countries, given the lack of public health and accessible sexual health infrastructure in some countries [88, 89]. Research is needed to explore the barriers and facilitators to seeking online sexual health information and support and to develop recommendations to overcome the barriers and enhance the facilitators for people from underserved populations in low-and-middle-income countries.

Regarding service specific operationalisation and implementation of our recommendations, intervention development guidance [86] highlights the centrality of involving of stakeholders to co-produce culturally appropriate real world intervention content and help secure their effective delivery, given their investment of time and resource. For improving online sexual health information and support for underserved populations, our BCW analysis has delivered the first step of stakeholder engagement, offering practical theory-and-evidence-based recommendations (i.e., Tables 4 and 5). However, for each recommendation, the operationalisation and implementation for specific settings requires thorough consideration by service providers and other relevant stakeholders (e.g., healthcare providers, public health organizations, NGOs,

government health departments, digital health platform developers, legal and regulatory bodies, funders and donors, educational institutions, and media and communication experts). Such stakeholder engagement should also consider affordability, practicability, efficacy, acceptability, side-effects, and equity [45] and address the operationalisation of our proposed recommendations to ensure final intervention content aligns with local circumstances and their wider legislative socio-cultural contexts. This could also involve the use of implementation science tools, such as Normalisation Process Theory [87, 90, 91], to ensure relevant issues of context are rigorously considered for maximum likelihood of success.

## Strengths

The PPIE activities with diverse representatives of underserved populations ensured the development of inclusive study materials that were fit for purpose. Moreover, our recruitment strategy of working with clinics and NGOs focussed on the recruitment of participants via in-person rather than online settings. This facilitated the recruitment of people who may struggle to use online sexual health services. In addition, use of the PROGRESSPlus framework enabled the recruitment of a diverse sample of people from a range of underserved populations whose needs are typically unmet by existing services and whose perspectives are often overlooked (e.g., minoritised sexual, ethnic, and religious groups, those living in the most deprived areas). The diversity of our sample supports the transferability of the findings to a wide range of people who may struggle to use existing online sexual health information and support services.

Further, our use of in-depth participant-led qualitative interviews and rigorous inductive thematic analysis conducted by trained and highly experienced qualitative researchers allowed for the collection of rich data and identification of important evidence-based barriers and facilitators to seeking online sexual health information and support. Moreover, the rigorous auditing of the thematic analysis by an experienced behavioural scientist and a larger inter-disciplinary team enabled the development of a comprehensive set of themes, many of which were found to be consistent with previous research. In line with Braun and Clarke [63, 92], we demonstrate a transparent and comprehensive account of the rigorous thematic analysis and subsequent BCW analysis which appropriately answered the research questions. In addition, use of the BCW enabled the development of theory informed recommendations for improvements to seeking online sexual health information and support. As the BCW draws on the cumulative learning from multiple theories of behaviour change [66], it enabled us to specify potentially useful intervention content beyond that suggested by our participants.

## Limitations

We sought to recruit a diverse sample of people from underserved populations with low digital literacy. However, while the sample was highly diverse from a range of underserved populations, it remains unclear whether the sample accurately represented people of lower digital literacy. The data we collected indicated that participants experienced difficulties using online sexual health services. However, a small majority self-reported as having high internet skills based on regular use of the internet to complete simple and familiar online tasks (e.g., social media, video streaming). Thus, a limitation of this study was that we did not use a validated measure of digital literacy. Although measures were available [93, 94], none were suitable, as they were too long and too involved, thus had the potential to limit participant engagement with the study. Research is needed to develop a validated short scale measure of digital literacy.

Further, no formal validation of the themes generated from the thematic analysis was conducted. While approaches are available, there is debate and no consensus regarding the best approach for this in qualitative research [95–97]. Additionally, in line with Braun and Clarke

[63, 92], no inter-rater reliability of the thematic analysis was conducted, due to the highly reflexive nature of the analysis. Instead, a rigorous audit of the thematic analysis was conducted by a behavioural science expert and a broad inter-disciplinary team. However, the consistency of the barrier and facilitator themes identified in this study with previous research does suggest the reliability of the study and validity of the findings.

Moreover, the barriers and facilitators and subsequent recommendations presented here are based on the specific experiences of the participants recruited for this study and may not be generalisable across all underserved populations. However, the consistency in our barrier and facilitator findings with previous research with more specific populations suggests that the results of this study and the recommendations generated are transferrable and relevant for a broad range of underserved groups in the national settings mentioned above. Nonetheless, further research is needed to validate the findings in other contexts and confirm their applicability across diverse underserved populations. Lastly, as this study used the PROGRESSPlus framework for purposive sampling of people from diverse underserved populations, some specific demographic factors were not captured, such as urban/rural location and distance to sexual health services. Therefore, further research is needed for such underserved populations that were not included in this study.

## Conclusion

This study identified key barriers and facilitators to seeking online sexual health information and support among a diverse sample of underserved populations. Our BCW analyses then suggested an array of potentially useful changes that could be made to reduce barriers and enhance facilitators to passing through this digital doorway and subsequently increase access to wider in-person and online sexual healthcare. Overall, our recommendations focus on adding to existing services in ways that enable used of them and educate, persuade, and offer accessible training to those from underserved populations. Recommendations include educate about the existence of online services that provide sexual health information and support; persuade people about their credibility; provide training, such as step-by-step instructions on how to seek information online and use online support and appraise the credibility of online information and support services; model peers successfully seeking information and using support services online; and enable use of them by ensuring they are inclusive and simple to use, including provision of providing a glossary of terms to assist with spelling for searches and communication and step-by-step instructions on how to seek information online and use online support. Ultimately, while online sexual health provision has the potential to extend access to healthcare for some, addressing the needs of underserved population outlined here is crucial to facilitate access to through the digital doorway to sexual healthcare and prevent the widening of health inequalities.

## Supporting information

**S1 Fig. An overview of seven interrelated elements of care within the behavioural system of accessing and using online sexual healthcare.** Thick yellow arrows indicate a non-sequential order, where the elements of care can occur in any order, e.g., getting sexual health information online can occur before or getting sexual health support online. Thick blue arrows indicate a sequential order, where a later element of care cannot precede an earlier domain, e.g., getting STI test result online must occur after getting a postal STI/BBV self-sampling kit online. Thin yellow arrows indicate that getting sexual health information or support online can occur at any point in the STI self-sampling and treatment pathway.
(TIF)

**S2 Fig. Second visual aid for online sexual health support seeking provided to every participant prior to the interview.**
(TIF)

**S1 Table. A list of examples of online sexual health information and support services.**
(DOCX)

**S2 Table. PROGRESSPlus informed target sampling frame developed prior to data collection and final sample targets met.** [a]SES = Socio-economic Status. [b]IMD = Index of Multiple Deprivation [70]. [c]Bold = Target met or exceeded.
(DOCX)

**S3 Table. Socio-economic demographics and screening survey.**
(DOCX)

**S4 Table. Interview topic guide.**
(DOCX)

**S5 Table. Sexual health resources information provided to each participant post interview.**
(DOCX)

**S6 Table. Participant self-reported skills and experience using the internet.** [a]Participant demographics for one participant were not obtained, table includes demographics for n = 34, except where participants did not wish to answer the question. Percentages are calculated for N = 35.
(DOCX)

**S7 Table. Intervention Functions and their definitions.** Adapted from Michie et al. [45].
(DOCX)

## Acknowledgments

We would like to thank the participants who took part in the study. We are also grateful to the following sexual health services and organisations for invaluable assistance with the study. We would like to thank Castle Circus Health Centre (Torbay & South Devon NHS Foundation Trust), Devon Sexual Health–Exeter & Barnstaple (Northern Devon NHS Foundation Trust), Mortimer Market Centre, Archway, Barnet and Buryfields Clinics (Central and North West London NHS Foundation Trust), Sandyford Clinic (Greater Glasgow & Clyde), and Sexual Health Sheffield (Sheffield Hospitals NHS Foundation Trust), for advertising the study to support with recruitment. We would also like to thank Hidayah LGBT and get2gether, for publicising the study to help with recruitment.

## Author Contributions

**Conceptualization:** Julie McLeod, Jennifer MacDonald, Paul Flowers.

**Formal analysis:** Julie McLeod, Paul Flowers.

**Funding acquisition:** Claudia S. Estcourt, Jo Gibbs, Melvina Woode Owusu, Ann Blandford, Paul Flowers.

**Investigation:** Julie McLeod.

**Methodology:** Jennifer MacDonald.

**Project administration:** Melvina Woode Owusu.

**Resources:** Julie McLeod, Jennifer MacDonald, Fiona Mapp, Paul Flowers.

**Supervision:** Claudia S. Estcourt, Paul Flowers.

**Visualization:** Julie McLeod, Paul Flowers.

**Writing – original draft:** Julie McLeod, Paul Flowers.

**Writing – review & editing:** Julie McLeod, Claudia S. Estcourt, Jennifer MacDonald, Jo Gibbs, Melvina Woode Owusu, Fiona Mapp, Nuria Gallego Marquez, Amelia McInnes-Dean, John M. Saunders, Ann Blandford, Paul Flowers.

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
