## [Decision Letter · Decision Letter 0]

16 Aug 2024

PONE-D-24-30763Opening the digital doorway to sexual healthcare: Recommendations from a Behaviour Change Wheel analysis of barriers and facilitators to seeking online sexual health information and support among underserved populationsPLOS ONE

Dear Dr. McLeod,

Thank you for submitting your manuscript to PLOS ONE. After careful consideration, we feel that it has merit but does not fully meet PLOS ONE’s publication criteria as it currently stands. Therefore, we invite you to submit a revised version of the manuscript that addresses the points raised during the review process.

We look forward to receiving your revised manuscript.

Kind regards,

Rabie Adel El Arab

Academic Editor

PLOS ONE

Journal Requirements:

This research is part of the SEQUENCE Digital programme funded by the National Institute for Health Research (https://www.sequencedigital.org.uk/about)

4. In this instance it seems there may be acceptable restrictions in place that prevent the public sharing of your minimal data. However, in line with our goal of ensuring long-term data availability to all interested researchers, PLOS’ Data Policy states that authors cannot be the sole named individuals responsible for ensuring data access (http://journals.plos.org/plosone/s/data-availability#loc-acceptable-data-sharing-methods).

5. We note that this data set consists of interview transcripts. Can you please confirm that all participants gave consent for interview transcript to be published?

If they DID provide consent for these transcripts to be published, please also confirm that the transcripts do not contain any potentially identifying information (or let us know if the participants consented to having their personal details published and made publicly available). We consider the following details to be identifying information:

- Names, nicknames, and initials

- Age more specific than round numbers

- GPS coordinates, physical addresses, IP addresses, email addresses

- Information in small sample sizes (e.g. 40 students from X class in X year at X university)

- Specific dates (e.g. visit dates, interview dates)

- ID numbers

Or, if the participants DID NOT provide consent for these transcripts to be published:

- Provide a de-identified version of the data or excerpts of interview responses

- Provide information regarding how these transcripts can be accessed by researchers who meet the criteria for access to confidential data, including:

a) the grounds for restriction

b) the name of the ethics committee, Institutional Review Board, or third-party organization that is imposing sharing restrictions on the data

c) a non-author, institutional point of contact that is able to field data access queries, in the interest of maintaining long-term data accessibility.

d) Any relevant data set names, URLs, DOIs, etc. that an independent researcher would need in order to request your minimal data set.

For further information on sharing data that contains sensitive participant information, please see: https://journals.plos.org/plosone/s/data-availability#loc-human-research-participant-data-and-other-sensitive-data

If there are ethical, legal, or third-party restrictions upon your dataset, you must provide all of the following details (https://journals.plos.org/plosone/s/data-availability#loc-acceptable-data-access-restrictions):

a. A complete description of the dataset

b. The nature of the restrictions upon the data (ethical, legal, or owned by a third party) and the reasoning behind them

c. The full name of the body imposing the restrictions upon your dataset (ethics committee, institution, data access committee, etc)

d. If the data are owned by a third party, confirmation of whether the authors received any special privileges in accessing the data that other researchers would not have

e. Direct, non-author contact information (preferably email) for the body imposing the restrictions upon the data, to which data access requests can be sent

6. Please amend either the abstract on the online submission form (via Edit Submission) or the abstract in the manuscript so that they are identical.

**Additional Editor Comments:**

Dear Authors,

Thank you for submitting your manuscript titled “Opening the digital doorway to sexual healthcare: Recommendations from a Behaviour Change Wheel analysis of barriers and facilitators to seeking online sexual health information and support among underserved populations” for consideration. Your work addresses a crucial issue, and the findings have the potential to make a significant impact. However, I recommend the following specific revisions:

1. Enhance the Global Relevance

Action: Expand the discussion to address how the findings might be applicable in non-UK contexts, particularly in low- and middle-income countries.

Specific Examples: Include a paragraph in the discussion section that explores how the barriers and facilitators identified in your study could manifest differently in healthcare systems outside the UK, especially where digital infrastructure may be less developed. Suggest possible adaptations of your recommendations for these settings.

2. Address Overestimation and Underestimation of Data

Action: Revisit the discussion section to ensure that the recommendations are appropriately scaled to the study’s sample size and qualitative nature. Explicitly state the limitations in generalizing the findings across all underserved populations.

Specific Examples: When discussing recommendations like improving digital literacy or providing video/audio options, clarify that these are based on the specific experiences of your participants and may need further validation in other contexts. You could add language such as, “While these recommendations are grounded in the data from our study sample, further research is needed to confirm their applicability across diverse underserved populations.”

3. Explicitly Acknowledge the Study’s Limitations

Action: Include a dedicated subsection on limitations within the discussion to transparently address the study’s constraints.

4. Provide a More Nuanced Data Interpretation

Action: Refine the interpretation of your findings to avoid broad generalizations. Highlight the complexity and variability of barriers and facilitators across different subgroups within the underserved populations.

Specific Examples: In the results section, when discussing barriers such as privacy concerns, acknowledge that these concerns might vary significantly depending on cultural background, age, and digital literacy levels. Consider adding a sentence like, “Privacy concerns were significant among our sample; however, these concerns may differ in intensity and nature among other subgroups not represented in this study.”

5. Detail Methodological Rigor

Action: Provide more transparency regarding the thematic analysis process. Detail how themes were validated and whether any checks for inter-rater reliability were conducted.

Specific Examples: In the methods section, add a brief description of the coding process and whether multiple researchers were involved in analyzing the data. If inter-rater reliability was not assessed, consider acknowledging this as a limitation and suggest that future research could incorporate this step for enhanced rigor.

The manuscript does not mention whether data saturation was achieved. Qualitative research does not aim for generalizability in the same way as quantitative studies, but transferability (the applicability of findings to other contexts) is important. The manuscript does not sufficiently discuss how the findings might transfer to other underserved populations or settings.

6. Discuss Policy and Practice Implications

Action: Expand the discussion on the practical implications of your findings, particularly focusing on how your recommendations could be implemented in real-world settings.

Specific Examples: For each major recommendation (e.g., improving digital literacy, increasing inclusivity), include a brief discussion of the steps needed for implementation at various levels. For instance, “To implement our recommendation of increasing digital literacy, local health authorities could collaborate with community organizations to deliver targeted training programs that are culturally and linguistically appropriate.”

I look forward to reviewing the revised manuscript

Best regards,

Reviewers' comments:

Reviewer's Responses to Questions

**Comments to the Author**

1. Is the manuscript technically sound, and do the data support the conclusions?

Reviewer #1: Partly

Reviewer #2: Yes

2. Has the statistical analysis been performed appropriately and rigorously? 

Reviewer #1: N/A

Reviewer #2: Yes

3. Have the authors made all data underlying the findings in their manuscript fully available?

Reviewer #1: Yes

Reviewer #2: Yes

4. Is the manuscript presented in an intelligible fashion and written in standard English?

Reviewer #1: Yes

Reviewer #2: Yes

5. Review Comments to the Author

**Reviewer #1:** 1. Which of the qualitative methods has been used?

2. Add interview questions

3. Reference should be given for the steps of data reduction

4. The criteria used to validate the findings should be added.

5. In a table, the process of abstraction should be mentioned (for a one them from the interview text

6.

You can also use the following Qualitative articles

1. Taghipour, A., Karimi, F., Latifnejad Roudsari, R., Mazlom, S. Coping Strategies of Women Following the Diagnosis of Infertility in Their Spouses: A Qualitative Study. Evidence Based Care, 2020; 10(1): 15-24. doi: 10.22038/ebcj.2020.42136.2120

2. Taghipour A, Karimi FZ, Roudsari RL. Exploring iranian women’s perceptions and experiences of their spouses’ behavior towards male factor infertility: A qualitative study. Current Women's Health Reviews. 2020 Feb 1;16(1):60-8.

**Reviewer #2:** Overall this is a well written manuscript with sound qualitative methodology. The authors aim to identify barriers and facilitators for seeking online sexual health information and support, and use their findings from semi-structured interviews to make recommendations to support underserved populations.

I do feel that the findings from this study add to the medical literature to advance access to online sexual health services for vulnerable or marginalized populations.

The main limitation is with the participants-- while diverse in age, ethnicity and education, such wide diversity may limit applicability of the findings to specific populations (ie young adults). Moreover, certain demographic factors were not included, such as urban/rural location, distance to medical services or establishment with a primary care physician. The authors state 51% lived in "most deprived areas of the UK". How is this defined-- average income? percentage falling below poverty level?

I would encourage the authors to acknowledge these limitations in the discussion section.

6. PLOS authors have the option to publish the peer review history of their article (what does this mean?). If published, this will include your full peer review and any attached files.

Reviewer #1: No

Reviewer #2: No

---

## [Author Response · Author response to Decision Letter 0]

15 Oct 2024

Dear Editor, 

We thank you and the reviewers for your quick and insightful review of our paper and the helpful feedback provided. 

We have addressed each of your and the reviewers’ comments, detailed in four tables in the submitted file labelled 'Response to Reviewers', and believe the paper has benefitted greatly from this. 

We look forward to your further review of this paper. 

Kind regards, 

Julie McLeod

---

## [Editor Report · Decision Letter 1]

21 Nov 2024

Opening the digital doorway to sexual healthcare: Recommendations from a Behaviour Change Wheel analysis of barriers and facilitators to seeking online sexual health information and support among underserved populations

PONE-D-24-30763R1

Dear Dr. McLeod

We’re pleased to inform you that your manuscript has been judged scientifically suitable for publication and will be formally accepted for publication once it meets all outstanding technical requirements.

Kind regards,

Ilhem Berrou, PhD

Academic Editor

PLOS ONE

Additional Editor Comments (optional):

Thank you for addressing the reviewers' comments. Please review the link posted on page 10 and here: (https://www.bct-taxonomy.com/) and provide a current reference if available.

---

## [Editor Report · Acceptance letter]

28 Nov 2024

PONE-D-24-30763R1 

PLOS ONE

Dear Dr. McLeod, 

I'm pleased to inform you that your manuscript has been deemed suitable for publication in PLOS ONE. Congratulations! Your manuscript is now being handed over to our production team.

Kind regards, 

on behalf of

Dr. Ilhem Berrou 

Academic Editor

PLOS ONE